# Hydrodeoxygenation of Levulinic Acid to γ-Valerolactone over Mesoporous Silica-Supported Cu-Ni Composite Catalysts

**DOI:** 10.3390/molecules27175383

**Published:** 2022-08-24

**Authors:** Margarita Popova, Ivalina Trendafilova, Manuela Oykova, Yavor Mitrev, Pavleta Shestakova, Magdolna R. Mihályi, Ágnes Szegedi

**Affiliations:** 1Institute of Organic Chemistry with Centre of Phytochemistry, Bulgarian Academy of Sciences, 1113 Sofia, Bulgaria; 2Laboratory of Inorganic Materials Chemistry, University of Namur, 5000 Namur, Belgium; 3Research Centre for Natural Sciences, Magyar Tudósok Körútja 2., 1117 Budapest, Hungary

**Keywords:** lignocellulosic biomass, levulinic acid, mesoporous silica composites, γ-valerolactone, gas-phase hydrogenation, solid state NMR

## Abstract

Monometallic (Cu, Ni) and bimetallic (Cu-Ni) catalysts supported on KIT-6 based mesoporous silica/zeolite composites were prepared using the wet impregnation method. The catalysts were characterized using X-ray powder diffraction, N_2_ physisorption, SEM, solid state NMR and H_2_-TPR methods. Finely dispersed NiO and CuO were detected after the decomposition of impregnating salt on the silica carrier. The formation of small fractions of ionic Ni^2+^ and/or Cu^2+^ species, interacting strongly with the silica supports, was found. The catalysts were studied in the gas-phase upgrading of lignocellulosic biomass-derived levulinic acid (LA) to γ-valerolactone (GVL). The bimetallic, CuNi-KIT-6 catalyst showed 100% LA conversion at 250 °C and atmospheric pressure. The high LA conversion and GVL yield can be attributed to the high specific surface area and finely dispersed Cu-Ni species in the catalyst. Furthermore, the catalyst also exhibited high stability after 24 h of reaction time with a GVL yield above 80% without any significant change in metal dispersion.

## 1. Introduction

The increased demand for energy and decrease in fossil fuel reserves, together with the high price of petrochemicals, has forced the economy to apply renewable energy resources [1,2,3,4,5]. Lignocellulosic biomass can be hydrolyzed into a mixture of cellulose, hemicellulose, and lignin. The hydrolysis of hemicellulose and cellulose results in the formation of C5 and C6 monosaccharides. Levulinic acid can be produced from lignocellulose via acid-catalyzed hydrolysis and can be further utilized as a platform molecule to produce valuable products including biofuel precursors such as γ-valerolactone [6,7]. GVL can be used as a solvent, fuel additive, and intermediate in the production of diverse value-added chemicals. Recently, a lot of investigations have been carried out on the catalytic hydrogenation of LA to GVL via homogeneous as well as heterogeneous catalysis [2,3]. Heterogeneous catalytic processes are economically and ecologically more relevant, offering advantages such as easy recovery and recycling. Typical catalysts for the liquid-phase hydrogenation of LA to GVL are supported noble metals (Ru, Ir, and Pd) or transition metals such as Co, Cu, and Ni [8,9,10,11,12,13,14,15,16,17,18,19]. Nickel catalysts supported on Al_2_O_3_, SiO_2_, TiO_2_, and ZrO_2_ have been extensively used as non-noble metal catalysts showing high activity during LA conversion and high selectivity toward GVL [9,10,14,15,18]. The presence of a second metal can influence the dispersion of two metals. The excellent catalytic activity could be ascribed to the synergistic effect between metals such as copper and nickel. Nickel promotion was shown to prevent the deactivation of copper–silica nanocomposite catalysts [20]. However, the metal content of the catalysts was high, being 80 wt. %. In addition, the formation of nickel–copper alloy might be important in preventing sintering and metal leaching in the developed Ni/Cu hydrotalcite catalysts [21]. Ni-based catalysts generally require high pressure for full LA conversion and high GVL selectivity. Ni/Al_2_O_3_ was reported to achieve 92% LA conversion with 100% GVL selectivity (4 h, 200 °C, and 50 bar H_2_) [18], while Fu et al. [19] achieved 100% LA conversion and 99.2% GVL selectivity with a Ni/Al_2_O_3_ catalyst, using dioxane as the solvent and slightly milder reaction conditions (180 °C, 2 h, 30 bar H_2_). Obregon et al. [22] reported 100% LA conversion with 91% GVL selectivity at 250 °C at 65 bar H_2_ for 2 h.

The conversion of LA to GVL in the liquid phase suffers from many disadvantages, such as working in high-pressure conditions, difficult purification via a reaction process, as well as safety issues and waste emission. In addition, active metal leaching is also significant in such harsh conditions. The vapor-phase hydrogenation of levulinic acid has great potential to overcome these disadvantages. The development of technology for the preparation of GVL via the hydrodeoxygenation of levulinic acid requires the optimization of reaction parameters (temperature, pressure, space velocity, and reaction time) and the type of utilized catalysts. Zeolites are widely employed industrial acid catalysts [1,2], but the sole presence of micropores often imposes significant limitations. Reagent and product diffusion in and out of the small pores of microporous carriers are seriously hindered for larger molecules. The application of a hierarchical pore system can solve some of these problems. Mesoporous silica/zeolite composite materials combine the interconnected network of micropores with structured mesopores and provide stability and acidity to catalysts. The amount and strength of acid sites in the composite material is determined by the type and crystallite size of the zeolite component. In a less crystallized form of zeolite nanoclusters, weaker acidic sites are formed, being advantageous because of hindered coke formation and the longer lifetime of the catalyst [23]. The balance between acidic and hydrogenation properties of the bifunctional catalyst is important for its hydrocarbon cracking activity and stability [24,25].

The introduction of low-cost transition metals supported on acid supports is a promising method for industrial applications requiring bifunctional catalysts (with metallic and acidic function). Based on our previous experience [24,25], nickel metallic species possess high activity for the toluene hydrogenation reaction and vapor phase transformation of LA to GVL. To increase the activity of LA to GVL transformation, bimetallic catalysts can be promising candidates.

In the present study, we describe the application of Ni- and/or Cu-functionalized mesoporous silica–zeolite composites for the vapor-phase hydrodeoxygenation of levulinic acid to γ-valerolactone at atmospheric pressure. Zeolite seeds were incorporated into a KIT-6-type mesoporous silica to provide acid sites in the composite. It is to be expected that larger pores of KIT-6 mesoporous silica facilitate the mass transfer of reagents and products in this type of reaction as well.

## 2. Results and Discussion

The textural properties of the prepared catalysts were characterized via XRD, nitrogen physisorption, and scanning electron microscopic methods. The low angle (left) and high angle (right) X-ray diffraction patterns of the parent and impregnated KIT-6 composite materials are shown in Figure 1. The pure KZ sample showed a sharp, intense peak around 0.88 2*θ*°, corresponding to the (211) plane of KIT-6 structure. Reflections with low intensities characteristic for the (220) and (420) planes indicated that the material had a highly ordered mesoporous structure with cubic Ia3d symmetry. Using metal functionalization, diffraction peaks with slightly declined intensities were observed, confirming that the metals were successfully loaded onto the mesoporous structure and that the symmetry of the support was not destroyed.

Higher angle reflections could be assigned to CuO and NiO crystalline phases. The signal around 45 2*θ*° belonged to the sample holder. Reflections of crystalline MFI-type zeolite could not be observed in XRD patterns, because the incorporated zeolite seeds were probably smaller than 5 nm, the detection limit of the XRD technique. The crystallite sizes of the metal oxide nanoparticles calculated using the Scherrer equation are presented in Table 1. In monometallic catalysts, nickel oxide was finely dispersed, whereas copper oxide appeared in bigger particles. In bimetallic formulations, NiO particles were somewhat bigger or equal-sized compared to those in monometallic ones; however, CuO crystallites became smaller. The presence of crystalline oxide phases is evidence that via the applied wet impregnation technique, the penetration of impregnating salts into the pore system was restricted, and a part of the metal oxide resided on the external surface of the support. On the other hand, it could be observed that the three-dimensional pore structure of KIT-6 restricted the formation of bigger oxide particles.

In levulinic acid conversion to GVL, catalysts were applied in reduced states via hydrogen pretreatment at 400 °C. Thus, the dispersion of metallic species was a more relevant parameter for catalytic activity. XRD patterns of ex situ reduced catalysts are shown in Figure 2, and crystallite sizes can be found in Table 1.

The formation of metallic copper and nickel could be observed in the reduced samples. In mono- and bimetallic copper catalysts, the particle size significantly increased compared to that in the oxidic state, from 16–22 nm to 60 nm. Finely dispersed nickel species remained well distributed by reduction; however, in bimetallic catalysts, the increase in particle size was higher compared to that in monometallic ones (Table 1).

SEM images of the composite materials (Figure 3A–D) revealed that the initial and metal-modified materials were composed of ~1 μm-sized particles. Metal-modified composite materials showed the agglomeration of particles.

The metal content of the samples determined via EDAX elemental analysis is summarized in Table 1. The aluminum content of the KZ catalyst was 0.7 wt. %. The copper content of the Cu-KZ catalyst corresponded to the nominal value; however, in other samples, it was somewhat lower. The nickel contents did not reach the loaded amount either, but they were similar to each other.

The presence of ZSM-5-type nanocrystals in the composite could be evidenced using FT-IR spectroscopy [23,26]. The spectrum of the KZ sample compared to well-crystallized ZSM-5 zeolite is shown in Appendix A. It showed an intensive band at about 545 cm^−1^, characteristic for the asymmetric stretching mode of double five ring (D5R) in the MFI structure. The same band could be observed in the composite sample, but with far less intensity, suggesting that the zeolite seeds were less crystallized. Most probably, they only belonged to a few unit cells representing short-range ordering, not observable using XRD analysis.

The parent and the modified materials were investigated using ^29^Si and ^27^Al NMR spectroscopy. The single-pulse ^29^Si spectra and relative fractions of the different Si structures obtained via the deconvolution and fitting the spectra with the DMFit software are presented in Figure 4 and Table 2, respectively.

The ^29^Si spectral behavior of both modified and parent silica systems was mainly governed by the KIT-6 structural properties. In all cases, three distinct resonances at −109, −100, and −91 ppm were detected, corresponding to Si(0Al), Si(1Al) + Si(OH), and probably Si(2OH) sites. Impregnation with metals did not significantly affect the overall contribution of the individual resonances, with only a slight increase in the quantity of the Si(0Al) units along with a decrease in the other components in the case of the bimetallic system. The relatively low Al content in all materials, however, suggested that the resonance at −100 ppm was mainly due to the presence of Si(OH) structural units, making the determination of the Si/Al ratio unreliable on the basis of ^29^Si NMR. To further confirm this, ^1^H→^29^Si CP MAS spectroscopy was used, which relies on the transfer of polarization from ^1^H to heteronuclei (^29^Si) via space dipole–dipolar interactions, thus allowing the selective enhancement of signals from ^29^Si sites with neighboring ^1^H nuclei. Figure 5 shows the ^1^H→^29^Si CP MAS spectra, overlaid with their ^29^Si single-pulse spectra for comparison. In all materials containing KIT-6 structures, the most pronounced enhancements were observed for the resonances at −91 ppm and −100 ppm in accord with the assignment of both peaks to Si-OH groups, while the moderate enhancement for the main resonance at −109 ppm could be attributed to the possible magnetization transfer from the neighboring bridged hydroxyl groups and/or coordinated water molecules. Although the ^1^H→^29^Si CP spectra could not be interpreted quantitatively, this result supports the presence of a significant fraction of Si-OH groups originating mainly from the KIT-6 component in all hybrid materials.

The coordination state of Al nuclei in the aluminosilica framework was investigated via ^27^Al NMR. (Figure 6). The low aluminum content posed some difficulties in the precise analysis of all the samples, but some general conclusions could be drawn on the basis of the ^27^Al spectra. In all cases, the main resonance corresponding to tetracoordinated framework Al sites (FAl) was centered at −56 ppm, and it was significantly broader compared to the pure zeolite (spectrum not shown; refer to Table 2 for half-widths of the signals). In the parent KZ material, a second resonance at around 0 ppm, characteristic for extra-framework Al (EFAl) or for three-coordinated framework Al species, was also observed, along with a broad resonance at around 35 ppm, which is associated with distorted symmetry of an Al environment [27,28,29]. Metal modification led to the disappearance of the 0 ppm resonance. The defected structure of zeolite seeds in the composite, or the more probable incorporation of Al to the mesophase, made it possible that the latter signal did not belong to EFAl AlO^+^ ions, but similarly to other zeolitic systems [30,31], to three-coordinated framework Al species. The incorporation of metal ions can change the coordination to a more symmetrical tetrahedral one by replacing a proton from a silanol group in the vicinity of an aluminum atom at the defect site. Ion-exchange with metal ions is rather characteristic for zeolitic materials; however, in our case, the formation of a mesoporous aluminosilicate phase was to be expected.

N_2_ adsorption/desorption isotherms of ZK composites are illustrated in Figure 7. ZK nanocomposites showed type-IV isotherm with an H1 hysteresis loop at high relative pressures, characteristic for mesoporous materials with pore sizes above 4 nm. The calculated textural parameters are summarized in Table 3. Functionalization with Cu/Ni resulted in decreased surface area and pore volume but not in decreased pore size. The decreases were more pronounced in bimetallic samples with higher metal contents. This can be explained by pore blocking, i.e., metal-oxide deposition inside the channels, despite the presence of metal oxides as a separate phase, as evidenced via XRD (Table 3). It seems that a part of Ni and/or Cu oxides could be found inside the internal space of the silica in an amorphous state, and the incorporation of metal ions into the silica lattice cannot be excluded either. The latter could also be assumed using NMR data showing the transformation of three-coordinated framework Al species into tetrahedral ones. The presence of a zeolite-like structure in the composite allowed for stronger interaction between the metal precursor and the zeolite clusters.

TPR experiments were performed in order to study the redox properties of the formed metal oxides (Figure 8). The reduction of copper oxides on KZ was characterized by one step at 190 °C. This effect could be due to the narrower particle size distribution of copper oxides on the support compared to the bulk one. Monometallic Ni samples showed two separated reduction peaks at around 260 and 370–380 °C, indicating different sizes of nickel oxide particles. Bimetallic modification led to a significant decrease in NiO reduction temperature, coalescing with that of copper oxide. It seemed that the vicinity of copper increased the reducibility of nickel. The simultaneous reduction of the metals and the absence of nickel reduction peaks at higher temperatures suggested the formation of an alloy phase on the silica support [32]. By calculating the amount of hydrogen consumption during reduction, we found that metal could can be reduced to a fully metallic state in all catalysts up to 400 °C.

The catalytic hydrodeoxygenation of LA to γ-valerolactone was studied at 200–250 °C reaction temperatures (Figure 9). GVL can be formed from LA via consecutive dehydration/hydrogenation reaction steps (Figure 1) [33,34,35,36]. If hydrogenation is the first step (Path II), then 4-hydroxypentanoic acid (HPA) is the intermediate product, which is dehydrocyclized to GVL. On path I, LA is first dehydrated to α-angelica lactone (AAL) intermediate and then hydrogenated to γ-valerolactone. On a bifunctional catalyst with hydrogenating and acid (Brønsted) functions, GVL is hydrogenated to pentanoic acid (PA) [36]. It was supposed that Brønsted acid sites initiate the scission of the (CH_3_C)-O bond in the GVL ring, and pentenoic acid is formed through a protonated intermediate [36].

Under the studied reaction conditions, we did not detect a 4-hydroxypentanoic acid intermediate, in accordance with the previous results obtained for Co/SiO_2_ [35]. Rather, an angelica lactone (AAL) intermediate was registered; therefore, based on the reaction products, the domination of reaction pathway I was supposed (Figure 1). Due to the higher hydrogenation rate at higher temperatures, the yield of α-angelica lactone decreased; moreover, it could not be found in the reaction product of the CuNi-KZ-6 catalyst. However, the higher reaction temperature also enhanced the further transformation of GVL to pentanoic acid in a hydrogenation step [36,37,38]. This effect was more pronounced in the presence of CuNi-KZ due to its higher metal content, easier reducibility, and enhanced hydrogenation activity.

The monometallic Ni-containing sample showed higher catalytic activity to GVL than the Cu-containing one at both reaction temperatures, also in connection with its stronger hydrogenation activity. Among all the studied samples, the bimetallic CuNi-KZ catalyst showed the highest conversion both at 200 °C and 250 °C. The optimal ratio between acidic and metallic active sites and structural peculiarities of the catalysts led to higher activity and selectivity in the hydroconversion of LA to GVL.

CuNi-KZ was not only the best-performing catalyst, but it also showed stable levulinic acid hydrogenation activity to GVL in 24 h on stream at 250 °C (Figure 10). The activity only decreased from 100% to 95%. Negligible changes were observed in the product selectivity, the yield of GVL decreased from 81.4% to 69.0%, and the yield of PA increased from 18.6% to 26.1%.

The physicochemical characterization of the spent CuNi-KZ catalyst did not show any change in the dispersion of metal nanoparticles (Appendix A). The presence of acid sites was important for the performance of the catalytic reaction, and they also ensured fine metal dispersion due to the strong interaction between metal oxide species and the support.

## 3. Conclusions

Monometallic (Cu, Ni) and bimetallic (Cu-Ni) catalysts supported on mesoporous silica/zeolite composites were successfully prepared. Catalysts were studied in lignocellulosic biomass-derived levulinic acid hydrodeoxygenation to GVL. Among the studied catalysts, CuNi-KZ showed the highest LA conversion (100%) and γ-valerolactone (GVL) yield (81%) at 250 °C. The high LA conversion and GVL yield could be attributed to the high specific surface area of the composite material and to the balance between the acidic and metallic Cu-Ni sites with high dispersion. Furthermore, the CuNi-KZ catalyst exhibited high stability after 24 h on stream, with a GVL yield above 80% and no change in the structure of the support and dispersion of metal nanoparticles.

## 4. Experimental Section

### 4.1. Materials

The chemicals used for synthesis of catalysts were tetraethyl orthosilicate (TEOS, 99%, Aldrich, Budapest, Hungary), tetrapropylammonium hydroxide (TPAOH, 20% in water, Sigma-Aldrich), aluminum isopropoxide (AlIP, 98%, Sigma-Aldrich, Budapest, Hungary), triblock copolymer pluronic P123 (EO20PO70EO20, MW = 5800 g/mole, Sigma-Aldrich), hydrochloric acid (HCl, 37% in water, J.T. Baker), ammonium hydroxide (25% of NH_3_ in water, Acros), and for the catalytic reaction, levulinic acid (98%, Aldrich). The modification of the KZ catalyst with transition metals was performed with Cu(II) acetylacetonate (Cu(O_2_C_5_H_7_)_2_, Aldrich) and nickel nitrate hexahydrate (Ni(NO_3_)_2_∙6H_2_O, Sigma-Aldrich).

### 4.2. Synthesis of Mesoporous Aluminosilica/Zeolites Composites

The synthesis of the silica/zeolite composite was performed in two stages [26]. In the first step, MFI-type zeolite seed solution was synthesized and pre-crystallized. Then, the remaining non-crystalline fraction of the precursor was transformed into an ordered mesoporous aluminosilicate with the help of a triblock copolymer template. The composition of the synthesis gel of the zeolite precursor was: 6.0 g of TEOS, 10.0 g of TPAOH, 2.0 g of distilled H_2_O, and 0.19 g of aluminum isopropoxide. The mixture was pre-crystallized at 90 °C for 24 h. 

For the synthesis of the KIT-6-based composite, 2.0 g of P123 was dissolved in a mixture of 72 g of distilled water and 4.0 g of 37 wt. % HCl with stirring at 35 °C. Then, 2.0 g of n-BuOH was added and stirred for 4 h until a completely clear solution was obtained. The zeolite precursor solution was added dropwise to the micelle-template-containing mixture, followed by aging at 40 °C for 24 h. Subsequently, the temperature was increased to 100 °C, and the mixture was further aged at the same temperature for 24 h under static conditions. The composition of the synthesis gel was the following: 1.0 TEOS: 0.016 Al_2_O_3_: 0.35 TPAOH: 0.012 P123: 1.42 HCl: 1.0 BuOH: 160 H_2_O. The white precipitate product was filtered without further washing and dried under vacuum at 100 °C overnight. The template was removed via calcination at 550 °C for 6 h. The obtained material was denoted as KZ.

### 4.3. Impregnation of Zeolite–Mesoporous Silica Composites with Nickel and/or Copper Salts

The wet impregnation technique with nickel and copper salts was applied for the loading of 10 wt. % metal for monometallic samples and 10 wt. % of each metal for bimetallic catalysts. The KZ support was dehydrated at 160 °C for 2 h before the impregnation procedure. Then, 0.5 g of KZ was mixed with 228.7 mg of Cu(II) acetylacetonate dissolved in 5 mL of chloroform or with 275.50 mg of Ni(NO_3_)_2_∙6H_2_O dissolved in 4 mL of ethanol at room temperature for the loading of 10 wt. % Ni or Cu, respectively. Samples were dried at 80 °C for 18 h. Precursor salts were decomposed in air at 400 °C for 3 h for all types of samples. The monometallic samples were designated as Cu-KZ and Ni-KZ. The bimetallic CuNi-KZ sample was prepared using a two-step procedure. In the first stage, the modification with Ni was performed followed by a second step with Cu. The obtained sample was designated as CuNi-KZ. The steps of catalysts preparation and the presumed structure of the mesoporous silica/zeolite composite is presented in Figure 2.

### 4.4. Characterization

X-ray powder diffraction patterns were recorded using a Philips PW 1810/3710 type (Bruker AXS Advanced X-ray Solutions GmbH, Karlsruhe, Germany) diffractometer applying monochromatized CuKα radiation (40 kV, 35 mA). Patterns were collected between 3 and 75°2*θ* with 0.02° step size for 1 s. The crystallite size of the metal oxides was determined using the Sherrer equation, evaluating the FWMH values of the oxide phases with a full profile fitting method.

Specific surface areas and pore volumes of the samples were determined from N_2_ physisorption isotherms collected at −196 °C using AUTOSORB iQ-C-MP-AG-AG (Quantachrome Instruments, Anton Paar brand, Boynton Beach, FL 33426, USA). Samples were pre-treated at 350 °C in a vacuum before nitrogen adsorption. Total pore volume was determined according to the Gurvich rule at a relative pressure of 0.98.

The temperature-programmed reduction–thermogravimetric analysis (TPR-TGA) investigations were performed using a STA449F5 Jupiter type instrument of NETZSCH Gerätebau GmbH (Gerätebau GmbH (Netzsch, Germany)). In a typical measurement, 20 mg of sample was placed in a microbalance crucible and heated in a flow of 5 vol. % H_2_ in Ar (100 cm^3^/min) up to 500 °C at a rate of 5 °C/min and a final hold-up of 1 h. Prior to the TPR experiments, the samples were treated in situ at 500 °C in air flow (10 °C/min) for 1 h.

The size and the morphology of the samples were studied with the SUPRA 35 VP scanning electron microscope (Carl Zeiss).

NMR spectra were recorded on a Bruker Avance II+ 600 NMR spectrometer (Bruker, Karlsruhe, Germany) operating at 600.01MHz ^1^H frequency (119.21MHz for ^29^Si, 156.34MHz for ^27^Al), using a 4 mm solid-state CP/MAS dual ^1^H/^31^P-^15^N probehead. The samples were packed in 4 mm rotors (Zr_2_O) and spun at a magic angle spinning (MAS) rate of 10 kHz for the measurement of ^29^Si spectra and at 14 kHz for the ^27^Al spectra. The quantitative ^29^Si NMR spectra were acquired with a single-pulse sequence, a 90° pulse length of 4.5 s, time domain data points of 3 K, and a spectrum width of 29 kHz, and 400 transients were accumulated with a relaxation delay of 120 s between each scan. The spectra were zero filled to 16 K data points and processed with an exponential window function (line broadening factor 10) before the Fourier transformation. The ^27^Al spectra were acquired with a single-pulse sequence, a 90° pulse length of 2.8 s, 128 K time domain data points, a spectrum width of 780 kHz, 1024 scans, and a relaxation delay of 0.5 s. All spectra were processed with an exponential window function (line broadening factor 10). The following experimental parameters were used for measuring the ^1^H→^29^Si cross-polarization MAS (CP MAS) spectra: a ^1^H excitation pulse of 3.6 s, contact time of 2 ms, a 5 s relaxation delay, more than 4000 scans, and an MAS rate of 10 kHz. The ^1^H SPINAL-64 decoupling scheme was used during the acquisition of CP experiments.

### 4.5. Catalytic Activity Measurements

Prior to the catalytic tests, samples were pretreated for 1 h in N_2_ flow at 400 °C. Levulinic acid hydrodeoxygenation was studied at atmospheric pressure using a fixed-bed flow reactor with hydrogen as the carrier gas (30 mL/min). In the reaction, a 50 mg sample (particle size 0.2–0.8 mm) was tested and diluted with 50 mg of glass beads of the same diameter, which were previously checked to be inactive. The reactor itself was a quartz tube with a 15 mm inner diameter, with the catalyst bed in the middle. A thermocouple was positioned in the catalyst bed for accurate temperature measurements. All gas lines of the apparatus were heated continuously to 150 °C in order to minimize the condensation of reactants and products on the tube walls. The hydrogen stream passed through a saturator filled with levulinic acid equilibrated at 80 °C. The reactants were fed into the reactor with a flow rate of 30 mL/min, and catalytic tests were carried out at 250 °C. The reaction steady state was established after 30 min at each temperature. On-line GC analysis of the reaction products was performed using NEXIS GC-2030 ATF (Shimadzu Group Company, Japan) with a 30 m HP-5MS capillary column. The stability of the most active sample was studied in the catalytic reaction with time on stream in 24 h. The applied mass balance was based on C-containing products and reactants. The conversion of LA (X_LA_) and the yield of products (Y_Xi_) were calculated as follows:

X_LA_ = (C_LA1_ − C_LA2_)/C_LA1_ × 100%

Y_Xi_ = C_LA2_ − ∑C_Xi-1_

In the equations, C_LA1_ is the mass fraction of LA in the inlet flow, C_LA2_ is the mass fraction of LA in the outlet flow, C_Xi_ is the mass fraction of the Xi product, and i is the products’ number. The carbon content of the feed and the reactor output showed a difference of less than 5% in each catalytic run.

## Data Availability

Not applicable.

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
