# Peer review of "Hydrodeoxygenation of Levulinic Acid to γ-Valerolactone over Mesoporous Silica-Supported Cu-Ni Composite Catalysts"

_molecules, 2022, doi:10.3390/molecules27175383_

Round 1

Reviewer 1 Report

The authors reported in the manuscript the development of mono/bimetallic catalysts immobilized onto zeolite-mesoporous silica to act as heterogeneous catalyst for the hydrodeoxygenation of levulinic acid to γ-valerolactone.

Overall, the manuscript is very well written, with a very detailed experimental part and the materials characterization as well as a strong discussion of the obtained results of the heterogeneous catalysis.

Nevertheless, minor revisions and modifications are necessary:

In the Introduction, is missing the literature state-of-the-art, regarding similar systems (mono and bimetallic acid/porous heterogeneous catalysts) to the conversion of levulinic acid to γ-valerolactone.

Page 2 Figure 1 and Figure SD1: Please correct the xx axis to 2θ (º). Also change this in the text.

Page 4 line 112: This size value corresponds to which material? The solid support or the metallic particles. Please clarify this.

Page 5 Figure 4: is missing the designation of xx axis.

Page 11 Experimental part: In the point 4.1 Materials are missing some used reagents (e.g. metals percursors). Please check and add.

After these modifications, I consider the article suitable to be published in Molecules.

Author Response

The authors thank the referees for the comprehensive review. The suggestions have been accepted and were taken into consideration. The needed corrections have been made and a corrected version of the manuscript has been submitted.

The authors reported in the manuscript the development of mono/bimetallic catalysts immobilized onto zeolite-mesoporous silica to act as heterogeneous catalyst for the hydrodeoxygenation of levulinic acid to γ-valerolactone. Overall, the manuscript is very well written, with a very detailed experimental part and the materials characterization as well as a strong discussion of the obtained results of the heterogeneous catalysis. Nevertheless, minor revisions and modifications are necessary:

Remark: In the Introduction, is missing the literature state-of-the-art, regarding similar systems (mono and bimetallic acid/porous heterogeneous catalysts) to the conversion of levulinic acid to γ-valerolactone.

Answer: The needed discussion was added.

Remark: Page 2 Figure 1 and Figure SD1: Please correct the xx axis to 2θ (º). Also change this in the text.

Answer: The requested changes have been made, also in the text.

Page 4 line 112: This size value corresponds to which material? The solid support or the metallic particles. Please clarify this.

Answer: The SEM data reveal the sizes of the composite material. Additional information was added in order to clarify the text.

Page 5 Figure 4: is missing the designation of xx axis.

Answer: We apologize for the mistake, the designation was added.

Page 11: Experimental part: In the point 4.1 Materials are missing some used reagents (e.g. metals percursors). Please check and add.

Answer: The needed information was added.

Reviewer 2 Report

The authors presented a complex catalyst to catalyze a complex reaction. The combination of Cu and Ni to gain a 100% conversion of LA is interesting. However, the authors shows a poor understanding of the results. Generally speaking, the several major issues impede the acceptance of the manuscript.

1)       What is the purpose of ZSM-5/mesoporous silica catalyst support used. Do the authors have the results of ZSM-5 or mesoporous silica as support?

2)       XRD patterns don’t show the 4- 40°range where the characteristic peaks of ZSM-5

3)       The catalyst structure is obscure, what is the relationship between KIT-6 and ZSM-5? what is the relationship between metal particles and the supports? Please draw a scheme.

4)       What is the main redox catalyst is, metals or metal ions or its combination?

5)       The catalytic result is poorly presented. At least the products distribution should be showed with a graph such as histogram rather than a simple table.

6)       Carbon balance is necessary.

7)       Catalyst stability test should be presented.

8)       The recipe for preparation of catalyst should show as molar ratio.

Author Response

The authors thank the referees for the comprehensive review. The suggestions have been accepted and were taken into consideration. The needed corrections have been made and a corrected version of the manuscript has been submitted.

The authors presented a complex catalyst to catalyze a complex reaction. The combination of Cu and Ni to gain a 100% conversion of LA is interesting. However, the authors show a poor understanding of the results. Generally speaking, the several major issues impede the acceptance of the manuscript.

1) What is the purpose of ZSM-5/mesoporous silica catalyst support used. Do the authors have the results of ZSM-5 or mesoporous silica as support?

Answer: We applied hybrid catalyst containing zeolite and mesoporous silica to take advantage of the beneficial properties of two components – acid sites of the ZSM-5 zeolite and the facilitated mass transfer in mesoporous structure of KIT-6. The obtained results are for the hybrid materials. Additional explanation was added in the MS.

2)       XRD patterns don’t show the 4- 40°range where the characteristic peaks of ZSM-5

Answer: The prepared composite samples do not contain crystalline ZSM-5, but rather zeolite seeds with small crystallite size, probably below 5 nm. Such small crystallites cannot be detected by XRD technique, the patterns at lower two theta values do not contain any reflection characteristic for ZSM-5. However, our NMR investigations proved the presence of zeolite in the structure, by detecting mainly tetrahedrally (75 wt. %) and trigonally coordinated (25 wt. %) Al species. The lack of crystalline ZSM-5 was mentioned in the XRD part of the manuscript.

3)       The catalyst structure is obscure, what is the relationship between KIT-6 and ZSM-5? what is the relationship between metal particles and the supports? Please draw a scheme.

Answer: The structure of the hybrid material consists of zeolite and mesoporous silica areas. There are a lot of papers in the literature based on such a type of materials. An additional explanation was added to the manuscript. The scheme with the main steps of the catalyst preparation was also contained.

4)       What is the main redox catalyst is, metals or metal ions or its combination?

Answer: The finely dispersed metal nanoparticles are the active sites together with acid sites in the catalysts.

5)       The catalytic result is poorly presented. At least the products distribution should be showed with a graph such as histogram rather than a simple table.

Answer: The results are presented in a 3D figure (Fig 9) and the discussion is revised.

6)       Carbon balance is necessary.

Answer: The carbon balance was described in the experimental part.

7)       Catalyst stability test should be presented.

Answer: The results of catalyst stability test are presented in Figure10 and discussed in the MS.

8)       The recipe for preparation of catalyst should show as molar ratio.

Answer: The recipe for the preparation of the catalyst is presented as molar ratio.

Round 2

Reviewer 2 Report

See the attachment

Author Response

The author made some efforts to improve the manuscript. However, there are still issue remains as follows.

The authors thank the referees for improving the quality of our manuscript. The suggestions have been taken into consideration, and the manuscript was further corrected.

The authors presented a complex catalyst to catalyze a complex reaction. The combination of Cu and Ni to gain a 100% conversion of LA is interesting. However, the authors show a poor understanding of the results. Generally speaking, the several major issues impede the acceptance of the manuscript.

1)        What is the purpose of ZSM-5/mesoporous silica catalyst support used. Do the authors have the results of ZSM-5 or mesoporous silica as support?

Answer: We applied hybrid catalyst containing zeolite and mesoporous silica to take advantage of the beneficial properties of two components - acid sites of the ZSM-5 zeolite and the facilitated mass transfer in mesoporous structure of KIT-6. The obtained results are for the hybrid materials. Additional explanation was added in the MS.

Comments: I suggest delete the ZSM-5 in the catalyst name since there is no evidence that ZSM-5 structure formed in the synthesis.

Answer: We agree with the reviewer, that our composite material contains less crystallized ZSM-5 zeolite particles, and our catalysts are rather similar to mesoporous silica alumina. However, ATR FT-IR spectra of the composite material showed a weak 545 cm-1 band characteristic for D5R rings in ZSM-5 structure (Supplementary data, FigS2), proving that there are some zeolite seeds in the structure. The name of the catalysts was changed to KZ, showing that the KIT-6 character of them is more emphasized. The manuscript was corrected according to the above facts.

2)        XRD patterns don’t show the 4- 40°range where the characteristic peaks of ZSM-5

Answer: The prepared composite samples do not contain crystalline ZSM-5, but rather zeolite seeds with small crystallite size, probably below 5 nm. Such small crystallites cannot be detected by XRD technique, the patterns at lower two theta values do not contain any reflection characteristic for ZSM-5. However, our NMR investigations proved the presence of zeolite in the structure, by detecting mainly tetrahedrally (75 wt. %) and trigonally coordinated (25 wt. %) Al species. The lack of crystalline ZSM-5 was mentioned in the XRD part of the manuscript.

Comments: Same with previous comment. NMR could not provide decisive conclusion of the presence of zeolite. If you perform the Al NMR analysis of mesoporous materisal, say KIT-6, you will find a similar results with yours.

Answer: The reviewer is right; NMR data alone does not evidence the presence of zeolite seeds in the structure. As mentioned before our ATR FT-IR results supported the presence of zeolite seeds, and the manuscript was corrected according to it.

3)        The catalyst structure is obscure, what is the relationship between KIT-6 and ZSM-5? what is the relationship between metal particles and the supports? Please draw a scheme.

Answer: The structure of the hybrid material consists of zeolite and mesoporous silica areas. There are a lot of papers in the literature based on such a type of materials. An additional explanation was added to the manuscript. The scheme with the main steps of the catalyst preparation was also contained.

Comment: You get misunderstand the means of scheme. Here attached an example.

Answer: We made a new scheme about the synthesis and probable structure of the prepared catalysts and inserted in experimental part.

5) The catalytic result is poorly presented. At least the products distribution should be showed with a graph such as histogram rather than a simple table.

Answer: The results are presented in a 3D figure (Fig 9) and the discussion is revised.

Comments : The histogram type is not good choice. Here I provide an example.

Answer: We have changed the conversion/yield figure to conversion/product distribution figure. The former trends (decreasing amount of AAL, and increasing ratio of PA) can be well observed in this type of figure, also.